# Lived Experiences of Mothering and Teaching during the Pandemic: A Narrative Inquiry on College Faculty Mothers in the Philippines

Alma Espartinez [1,2]

1  Theology-Philosophy Department, School of Multidisciplinary Studies, De La Salle–College of Saint Benilde, Manila 1004, Philippines; alma.espartinez@benilde.edu.ph or aesparti@providence.edu
2  Department of Philosophy, School of Arts & Sciences, Providence College, Providence, RI 02918, USA

**Abstract:** How do academic mothers navigate their embodied selves in a disembodied academic life? More particularly, how do mothers in Philippine Higher Education balance the demands of mothering and teaching during the pandemic? This qualitative study used a narrative inquiry approach involving in-depth interviews with academic mothers from various faculties and ranks at some Philippine Higher Education Institutions. This approach explored the complex and often contradictory discourses surrounding the tension between the polarizing models of the ideal caring mother and ideal academic, trying to excel in both roles during the pandemic. The research began with an overview by way of a literature review of the pre-pandemic mother academics. It then reflected on eight mother college professors who balanced their careers with childcare, some with adult care, as this pandemic amplified deeply ingrained traditional social norms that perpetuate social inequities. Finally, it concluded that the two domains—academy and family—remained inhospitable to professing mothers in the Philippines. This study proposed that care work should be valorized, work–family narratives normalized and mainstreamed, and public and educational policies that support mothering and teaching rethought.

**Keywords:** work–family conflict; COVID-19 pandemic; work–life balance; academic mothers; narrative inquiry; gendered norms

## 1. Introduction

The COVID-19 pandemic disrupted virtually all areas of our everyday lives. The work-from-home arrangement brought about by the stay-at-home directives has gendered dimensions (Adams et al. 1996; Boncori 2020; Padavic et al. 2019; Savigny 2014), and here is where women stated they were hurting the most. Academic mothers, in particular, disproportionately struggled when warped boundaries between home space and workplace, private and public life, embodied and disembodied existence, flesh and brain, paid and unpaid work, parenting and professing loom large as the pandemic extends longer than expected (Ajayi et al. 2020; Bahn et al. 2020; CohenMiller and Izekenova 2022; Galili 2020; King and Frederickson 2021; Ollilainen 2020). Studies were written on academic mothers who negotiated the demands of embodied experiences of motherhood and the disembodied expectations of the academe (Clarkson et al. 2021; Donaldson and Emes 2000; Halley et al. 2021). The academic mothers' multiple shifts—childcare, teaching duties, and household chores—were accompanied by anxiety about how they would navigate their lives' endless tasks (Boncori 2020; Deryugina et al. 2021).

This study engaged with the narratives of eight faculty mothers in Philippine Higher Education Institutions (HEIs) as they navigated complex and often contradictory discourses surrounding the tension between the polarizing models of the ideal mother and the ideal academic, exerting effort to excel in both during the pandemic. This study is limited to the study of heterosexual Filipina women married to their male spouses. It contributed to the

body of literature that voices maternal experiences in the field of female academic studies. More specifically, it addressed a gap in the literature on the lived experiences of academic mothers in the Philippine HEIs. I used the framework of Emmanuel Levinas's disinterested responsibility to epitomize the maternal sacrifice of academic mothers and David Whyte's concept of work–life balance to conceptualize the relationship between work and family.

This research appeared valuable for three reasons:

First, nothing has been written yet about the struggles of college academic mothers in their unique personal and professional circumstances in the Philippines during the COVID-19 pandemic. Most research studies on academic mothers' work–life balance were foreign-based: Ghana (Akuoko et al. 2021); United States (Beech et al. 2021); Kazakhstan, United Kingdom, Australia, Canada, South Korea, Lebanon, Ukraine, Ireland and Hungary (CohenMiller and Izekenova 2022); Italy (Minello et al. 2021); India (Bhende et al. 2020); New Zealand (Ryan et al. 2021); Kuala Lumpur (Mohamad and Despois 2022), among other studies. Articles abound on the various career stages of academic mothers (Briscoe-Palmer and Mattocks 2021; Fulweiler et al. 2021; Jones and Maguire 2021; Laudel and Gläser 2008; Merga and Mason 2020; Mirick and Wladkowski 2018; Pflaeger Young et al. 2021; Wilson et al. 2021). In a family-oriented Philippines, the pri-mary role of taking care of the home has been ascribed to women, and to men the duty to provide financially for the family. But when women are breadwinning, they also have to do the caring at home (Akanle and Nwaobiala 2020; Leupp 2020). The narrative stories of these academic mothers need hearing, now more than ever, as their burdens become complex, deeply ignored, and abysmally under-appreciated during the pandemic.

Second, the academic conditions under which women academics in the Philippine HEIs work provided the motivation for this research. Since the Philippines has a unique faculty teaching load arrangement (Pacaol 2021), which I further discuss in the literature review, the uniqueness of the situation in the Philippine academic setting may contribute to some variables that have not yet been richly explored, such as the heavy academic loads that exacerbate the already stressful lives of academic mothers, notwithstanding during the pandemic. The findings of this research expanded the care/career literature (Akuoko et al. 2021; Clarkson et al. 2021) and presented a unique tapestry of realities by examining the complexities of mothering with a teaching career in the Philippine context (Anonuevo 2000).

Third and last, this study can contribute to our understanding of the subaltern stories of these academic mothers, which can offer a platform to hear their voices and an essential perspective for scholars, school administrators, and policymakers to take a deeper look at the challenging roles of academic mothers in the Philippines as they perform their multiple functions—teacher, caregiver, and homemaker—and the prospects for adjusting university priorities, policies, and practices.

To this end, this research tried to capture the interplay between private and public, embodied (flesh) and disembodied (brain), home and work, and childcare and career in the life of mother academics in the Philippines. No clear analytical framework currently exists for constructing such an account. This gap in the literature grounds the significance of this study and its contributions to literature and practice.

## 2. Literature Review

The study articulated the complexities of motherhood and the teaching career of eight academic mothers in opposite-gendered marital relationships in Philippine tertiary education. Their narrative accounts were highlighted as they engaged with fleshy, embodied experiences of motherhood, sustaining the disembodied experiences of teaching. The conflicting expectations in their lives were foregrounded as these academic mothers tried to come to terms with, and made sense of, the unsettling changes in their personal and professional lives.

### 2.1. Women's Participation in the Workplace

Studies abound on how family and work affected the everyday life of women prepandemic (Ceci et al. 2014; Halpern and Cheung 2008; Hattery 2001; Maher 2013) and more so during these most difficult times of the current pandemic crisis (Reese et al. 2021; Tsouroufli 2020). Most faculty mothers' lives were deeply conflicted as they managed responsibilities in their personal life and professional work (Nawaz and McLaren 2016). Across the world, women—even those with professional jobs—performed more tasks and had less leisure time than their male partners (OECD 2020; Skinner et al. 2021). Care work (e.g., childcare and adult care) was a task assigned to women and it was usually non-monetized, unseen, and broadly viewed as not "real work." The consequence of combining career and motherhood was the motherhood penalty (Benard and Correll 2010; Chinchilla and León 2005). Increased participation by mothers in the workplace yielded a heavy burden on working mothers themselves (Veletsianos and Houlden 2020a). Professional women have remained imperiled, then and more now, in mothering and working.

### 2.2. Motherhood and the Academia

From the 11th to the 19th centuries, women were excluded from universities because they were viewed as less intellectually capable, less sensitive, and less capable of leadership than men (Benard and Correll 2010). Their services were of an auxiliary and charismatic nature. The academy, even leadership in business and politics, was reserved for men until women renegotiated their feminine roles in society and in the world (Selzer and Robles 2019; Zheng et al. 2018). Women started schooling and achieved their own academic degrees and eventually got employed in universities, although they were still underrepresented for one reason: they began having children (Armenti 2004; Moeke-Pickering et al. 2020). When mothers became academics and academics became mothers, the tug of war in, the continuous tension of, and the dialectical relationship between motherhood and teacherhood, the pulling on opposite ends of the rope, spelled indescribable stress and pressure for these academic mothers (Burk et al. 2020; Clarkson et al. 2021; Stead et al. 2021; Young 2015). Their embodied maternal experiences disrupting the disembodied academic work (Ollilainen 2020) have largely remained inadequately theorized (Bell and Sinclair 2014; Gabster et al. 2020) in career–motherhood studies. As a highly embodied reality, motherhood rendered an academic woman vulnerable and stigmatized in the workplace (Bassett 2005; Huopalainen and Satama 2019; Knowles et al. 2009) as she is perceived to have her primary loyalty at home. They published less and suffered reduced research productivity (Armenti 2004; Minello 2020; Minello et al. 2021; Mohamad and Despois 2022; Orchard et al. 2022). Further, research studies suggested that the failure to renegotiate academic and family responsibilities was a critical factor in persuading women to opt out of highly disembodied academia (Armenti 2004; Boushey 2005; McKie et al. 2013; Plotnikof et al. 2020; Stone 2008). This only showed the multiple challenges an academic mother experiences every day in her embodied maternal life and her disembodied teaching career.

### 2.3. Academic Mothers during the Pandemic

The demands on academic mothers pre-pandemic were already challenging and overwhelming, and their heavy workloads even worsened during the pandemic. The effect of the current pandemic on college faculty mothers was huge and unequal within and across countries. It changed their academic priorities in favor of their motherhood duties (Minello et al. 2021), highlighting the cumulative impact of the traditional distribution of gender roles in the family and the gendered view of work–family conflict (Santos and Cardoso 2008). Many academic mothers experienced unique challenges to scholarly productivity during the COVID-19 pandemic (CohenMiller and Izekenova 2022; Deryugina et al. 2021; King and Frederickson 2021; Myers et al. 2020; Pinho-Gomes et al. 2020) which is called the pandemic penalty (King and Frederickson 2021), adding to the motherhood penalty charged to them pre-pandemic. Unlike most female academics, male academics usually

prioritize their careers over domestic responsibilities and child care; they knew that their female partners would take on the tasks of doing household activities (Del Boca et al. 2020).

Due to the lockdown, academic mothers carried the added housework and childcare burden in the work-from-home arrangement. The deciding factor remained: whoever has lower income and flexibility stays home and takes care of the domicile duties (Bonacini et al. 2021; Briscoe-Palmer and Mattocks 2021). Given this pandemic scenario, how do academic mothers balance their career, childcare, and chores? In particular, how do college faculty mothers in the Philippines fare? How do they balance the challenges of contemporary motherhood and teaching in the academe during the pandemic?

### 2.4. Academic Mothers in the Philippines

Broadly speaking, I am interested in listening to the stories of academic mothers negotiating their faculty work and family care. Specifically, I find the Philippines an interesting perspective for working mothers in the academe. The gender situation in the Philippines views women as homemakers subordinate to men, making them either marginalized or discriminated against (Anonuevo 2000). Moreover, Philippine HEIs remained a gendered institution (Anonuevo 2000; Lao 2017; Evangelista 2017; Tarrayo et al. 2021). Academic mothers were assigned to nurture and mentor their students, mirroring their work in the home, with less emphasis on writing scientific research. Consider, also, the 18–24 teaching units college faculty members in the Philippines carry per semester (Pacaol 2021; Saliendra 2018; Tancinco 2016). While other foreign universities assign their faculty only 12 teaching units at most with 7–25 students in a class (Blatchford and Russell 2020; Chingos and Whitehurst 2011), the Philippine classroom is populated with 35–50, some even with 60, students with professors teaching eight classes per week with as much as three to four-course preparations (Saliendra 2018). As of this writing, no study has been conducted yet about the challenging teaching conditions of academic mothers in the Philippine HEIs during the pandemic. The attempt to make an account of Philippine college faculty mothers makes this study relevant research for academic women studies.

### 3. Research Questions

This study attempted to answer the following questions: (1) What are the challenges college academic mothers in Philippine Higher Education Institutions experience in navigating their embodied maternal lives in a disembodied academic world? More particularly, (2) how do college faculty mothers in the Philippines balance the challenges of contemporary motherhood and academia during the pandemic?

### 4. Methodology

My overall aim was to probe the multiple challenges of academic mothers in Philippine HEIs by foregrounding the stories of their lived personal and professional experiences during the pandemic. This research used the narrative inquiry methodology described by Clandinin (2006) as a rich qualitative methodology that studies participants' lived experiences. Narrative inquiry allows the participants to narrate their life stories and the researcher to "experience their experience" (Clandinin and Connelly 2004) while finding patterns and meanings emerging in these stories. Crucial to this approach is foregrounding the stories narrated by these academic mothers as I entered into the lived journey of their embodied (maternal) and disembodied (academic) lives. I used narrative inquiry as a platform where academic mothers described their experiences in their own stories and attempted to construct a counternarrative to deconstruct the dominant male narratives. Guided by the interpretivist framing, I explored inductively the lived experiences of the academic mothers in the Philippines by interpreting both manifest and latent content in their narratives to capture the essence of their stories. As a narrative inquirer, I studied my participants' stories through which I "seek ways of enriching and transforming that experience for themselves and others" (Clandinin 2006, p. 42), with the hope, in this research, of shifting academic motherhood at the margins toward the forefront (Bassett 2005).

This research used in-depth interviews with eight faculty mothers in Philippine HEIs. The rationale for the size is arbitrary as there is no specific size range in a narrative inquiry. I included participants at different stages in their academic careers to obtain a range of experiences that is more representative of academic motherhood in the Philippines. In the in-depth interviews, the life story interview method was applied. I documented the participant's personal stories by drawing multi-faceted histories as they reminisced about the past pre-pandemic, narrated current experiences during the pandemic, and anticipated their future post-pandemic. Each interview had an average duration of one hour and was audiotaped, transcribed, and analyzed. All the information was treated under the interviewees' consent. The interviewees' anonymity was maintained in the strictest confidentiality. They signed an interview consent form to express consent for research participation.

### 4.1. Participants

Using purposive sampling, I selected heterosexual academic mothers who could "purposefully inform" (Creswell and Poth 2007) my understanding of college faculty mothers' experiences in the Philippines. The criteria for inclusion in this research are that participants have to be employed as college faculty (lecturer to professor) and have at least one young child (between 0 and 18 years of age). To ensure diversity of background, I included academic mothers from private and public schools with varying age brackets, career levels, and diverse personal circumstances.

Eight academic mothers from different parts of the Philippines participated in this study through an initial email containing brief information about what the study aimed to achieve. Some of these college faculty are friends, others acquaintances, and the rest were recommended by administrators or colleagues from various academic institutions.

In my description of these academic mothers' experiences, I used the label MoM (which stands for Mothers of Multiples), numbered 1 to 8, to protect their identity. Confidentiality was guaranteed by not linking research findings to name or identity. The participants, likewise, knew that they could withdraw at any time from the study—at will. They were not compelled to participate in the research; no physical or mental discomfort was posed. No compensation was offered except for payment for their internet data usage during the interview.

### 4.2. Data Collection

This study employed an online survey using questionnaires with the informed consent of the invited participants. I emailed the participants the survey via a Google Form, which they answered between 15 December 2021 and 15 January 2022. The survey contained socio-demographic information and questions intended to reveal factors impacting the academic mothers' work-from-home arrangements in Philippine HEIs during the pandemic.

The in-depth interview questions were sent to the interviewees before the actual interview session. The interview is semi-structured to allow the interviewees to comfortably elaborate on general and essential themes, while the rest of the questions were asked spontaneously for probing to generate further explanations from research participants. I conducted the in-depth virtual interview using the Zoom video-conferencing app (which was the most convenient for all the participants) from 16 January to 10 February 2022, starting with some broad questions and then proceeding to specific questions on personal background and circumstances, work and home situation pre-pandemic and during the pandemic, challenges brought by warped boundaries of home and work spaces, and academic career trajectories.

### 4.3. Data Analysis

The qualitative interviews were recorded and transcribed (Halcomb and Davidson 2006). From the interview data, I began to analyze the narratives thematically using thematic analysis (Kiger and Varpio 2020), examining interview and conversation tran-

scripts, even going back and forth between them, purposely trailing and clarifying how the participants' interwoven stories painted a unique picture of the challenges of academic mothers in the Philippine academy. Then, I initiated coding by selecting key demographic characteristics of the personal circumstances of my interviewees (Saldana 2021).

Following the steps of thematic analysis "designed to search for common or shared meaning", as discussed by Kiger and Varpio (2020, p. 2) and adopted from Braun and Clarke (2006), I read the responses several times regarding the greatest struggles experienced as academic mothers during the pandemic. I then searched for, reviewed, defined, and named the themes generated from the "patterned response or meaning" (Braun and Clarke 2006, p. 82) derived from the interview data that informed the research questions. To develop trustworthiness, credibility, and validity, I shared my narrative interpretations with my participants to ensure their narrated stories matched my interpretation (Tobin and Begley 2004). The participants' feedback or member checking (Lincoln and Guba 1985) generated new insights; the participants felt empowered, involved, and committed to safeguarding the validity of the findings, making them active participants in my study. The audit trail for this narrative inquiry included audio-video recordings, interview transcripts, interview guides, a list of interviewees, and my personal notes.

## 5. Results of the Study

The participants consisted of eight heterosexual academic mothers in the Philippine HEIs with ages ranging between 35 and 55 years old. All of them have at least two children with ages 28 and below. Six work full-time and have tenure-track positions; two are adjunct faculty. Five are assistant professors, two instructors and one associate professor. Six hold administrative posts: one is a Chair of her department; another is a head librarian; and four are unit coordinators. One academic mother is happy with her husband and her two children.

Gifted with two intelligent children and a loving husband, I am a happy wife and contented mother. I am married to a husband who understands my work and my children who support the things I do. My world revolves around them. That's happiness for me. —MoM$_7$

Another academic mother has two children. She mentioned that life was more challenging during the pandemic, considering the increasing expenses she has been incurring for her medication, not to mention the diaper and milk for her toddler.

I am in my forties with 2-year-old and 10-year-old girls. I work in one Catholic college in Manila. My husband is a high school teacher. I have been looking for a full-time job, but it might be impossible to find one during the pandemic. Currently, I am undergoing dialysis for my kidney problem. —MoM$_6$

Two participants are single mothers with two school-age children each. Here is what they said about single parenting during the pandemic:

I am a proud single mother with two wonderful children whom I have raised singly and proudly for five years now since my husband—a gambler, womanizer, and drug addict—abandoned me for another woman. I was left in tatters, especially now while in the pandemic. It hurt my ego. My deep faith in God and my single motherhood are the main intersections of my identity that shape my commitment to see things through. I will earn my doctorate soon—my sweetest revenge. —MoM$_2$

I lost my husband to another woman because he couldn't stand my being an accomplished person. He is a weakling. He is in another country with his other woman while I am here fending for my two children. This pandemic is horrible. —MoM$_4$

An academic mother who is a college administrator has been taking care of her paralyzed husband who, despite his being wheelchair-bound, has never stopped being "bossy" to his breadwinning wife:

> I'm a full-time academic mother married to a husband who got paralyzed in 2018. I am the breadwinner in the family who is an adult caring for a hegemonic husband. I have three children—all girls. While they are all working, I don't depend on them financially. I work for my husband's medication. —$MoM_1$

From the in-depth interviews with these academic mothers, I identified two broad themes related to their lived experiences: *changing challenges* that described the trials they encountered while dealing with the COVID-19 pandemic and *championed choices* that made them transcend life's vicissitudes despite the debilitating impact of the pandemic on their personal and professional lives.

*5.1. Theme 1: Changing Challenges*

Three sub-themes were identified in this theme: (i) shifting gender roles; (ii) shifting grounds, and (iii) shifting fears. This study tried to capture the complexities of the lives of these academic mothers as they navigated and negotiated the gendered spaces of academia and family during the pandemic. The challenges they experienced as academic mothers seemed not to have subsided before and during the pandemic; they just kept changing. This study was my attempt to foreground their lived experiences, hoping to move my audience from the spot of spectator to witness.

5.1.1. Shifting Gender Roles: Female Breadwinning, Male Breadlosing

The first sub-theme showed how in the family-centered and patriarchal Philippines, men are expected to be the breadwinners and women the homemakers. This perception of gender roles largely remained and rarely reversed. However, in the case of the academic mothers in this study, six of the eight academic women interviewed earn more than their husbands, and two have husbands not earning at all before and during the pandemic. Practically, all of them are breadwinners. By breadwinner, I mean the sole provider or the primary income earner of income for dual-career couples. During the pandemic, some husbands lost their jobs, living expenses went up, and some family members were hospitalized due to COVID-19 infection. The escalating expenses contributed to their financial woes, especially for those whose husbands were out of work.

> My husband lost his job during the pandemic; he was jobless for a year and four months. I am the only one working. I stopped enrolling in graduate school. I gave up the purchase of a piece of land I bought. I couldn't manage the monthly amortization anymore. —$MoM_3$

> I am a 45-year-old mother married to a Palestinian-Jordanian, and we have four children. I am a professor, writer, emcee, comedienne at times, and binge crier most of the time. Since the pandemic, my husband has not found a job to replace the income lost. Beginning this year 2022, all my children caught symptoms of the omicron variant. We don't even have any money to buy medicine, much less have them swabbed. —$MoM_5$

> My husband does not have work. All my children are going to school. They used to go to private schools; now, they are all in public schools. I can no longer afford the high tuition fees in private schools. My children and I need to adjust our finances. —$MoM_5$

Due to the hard life in the Philippines, academic women were pressured to work full-time. Yet, while this was the case, they managed to tend to exhausting household chores and care for family members at home.

> I have more tasks to do—online classes, domestic chores and childcare, and adult care (I am taking care of my mother-in-law). On top of that, my husband expects

me to do the childcare. He takes care of my two-year daughter only when he's off from work or when he likes to. —MoM$_6$

I have 33 teaching units to pay for our daily needs. I even earn more than he does; I still take care of the household chores. I had taken on more childcare duties than before. I also pay for my mother's medication; she's an 80-year-old paralyzed woman. While my husband is working, I don't want to depend on him for my mother's expenses. —MoM$_8$

In North America and Europe, traditional breadwinning roles were challenged and, sometimes, reversed as more women join the workforce and even earn more than men (Myers and Demantas 2016). In the Philippines, academic mothers teach full-time at the university and work full-time at home. Even with stay-at-home dads, these academic mothers were still the ones cooking, doing the laundry, and taking care of the children. Their professional career did not excuse them from doing household chores and caring duties as these are the core feminine tasks required of them.

### 5.1.2. Shifting Grounds: Home–Work Spaces

The second sub-theme revealed how academic mothers found themselves crossing boundaries between work and home. Before the pandemic, these academic mothers knew where their commitment stood: it was dictated by the space they occupy. In school, they teach their students the best they could. At home, they changed diapers and washed dishes and dirty linens. A clear-cut boundary was set. Then came the pandemic, when work from home was the order of the day; home–work borders were warped and blurred. The shifting grounds confused them. Here are some of their complaints:

My paralyzed husband would go into tantrums suddenly at the slightest provocation while I was doing online classes and attending in-person academic meetings. Academic work and domestic duties become one bewildering mess. —MoM$_1$

Work-from-home gives me more demanding childcare and adult care. I take care of my 76-year-old diabetic father; I am a helpmate, caregiver, teacher, mother, and wife—all at once. Admittedly, this is not easy. I am so overwhelmed, and the work is overwhelming. —MoM$_3$

It wasn't easy tending to all the needs of my family. My life becomes a juggling act. I can still manage, though wishing my husband would be here to help me out. I am overwhelmed with caring, and nobody cares for me! —MoM$_3$

As academic mothers were highly-attuned to the sense of place, lacking it was highly disruptive in their lives, making them contemporary nomads. They were raging in wild anger and were close to snapping. Given these unforeseen and unsettling new challenges in their lives of mothering and teaching, they were frozen in their tracks. They were in meltdown mode, tail spinning into chaos. A plethora of anxieties was all over the place. This shifting academic and home spaces became a surprisingly unique challenge for even the most veteran professor and experienced mother. To be sure, the work-from-home arrangement is not so much something they grow out of as something they grow into.

### 5.1.3. Shifting Fears

The third sub-theme exposed how the academic mothers—confused and diffused—were gripped with fears. Two years and a half into the pandemic, with teeming responsibilities and back-to-back meetings, academic mothers grew exhausted as they moved from task to task to task.

College professors were negatively impacted by this immediate shift from face-to-face classes to virtual, technology-facilitated teaching (Dogra and Kaushal 2022). They were forced to migrate from in-person courses to online course designs requiring complex technology (Hodges and Fowler 2020; Rapanta et al. 2021; Veletsianos and Houlden 2020a, 2020b). This sudden academic shift brought anxiety, agitation, and tension among our

academic mothers, even exposing their technology illiteracy and inadequacy. The technostress of virtual teaching, endless online meetings, constant email exchanges, and unlimited group chats turned into a digital burden. While other younger ones could quickly adopt, the older ones were seriously technologically-challenged. Some of them even opted for early retirement due to the overwhelming challenges of online teaching (Skinner et al. 2021). Academic mothers were overwhelmed by complex technology; they were forced to tread a digital territory they were not familiar with. Two non-techie mothers narrated the following:

> I'm old to learning new ways of online teaching. Good thing my children are techies; they helped me navigate this new online mode of learning. If not for them, how will I survive this intricate technology! —MoM$_4$

> I'm 50+ already. Came the pandemic, I needed to navigate this fancy gadget and complicated LMS. I feel like I'm back to zero in teaching. I have heavier course preparation due to incorporating varied online teaching strategies. I wonder if my traditional teaching strategies still connect with the millennials. —MoM$_3$

Academic mothers transitioning to online learning needed to learn how students drop ideas into the chat box, quickly answer questions via online polls, and do small-group discussions in virtual breakout rooms. All these technologies were unfamiliar to them. While face-to-face teaching and learning was already a huge hurdle, the difficulty is more compounded by the transition to an online teaching mode.

> In our state universities, most of my students are not well-off. They don't have a stable wifi connection. How can you teach them well when they cannot even connect online? I have six classes with 60 students per class—360 students per semester! Yes, 360 students in one semester, all submitting their assignments online! Some take screenshots of their answers, and I have to read them one by one. —MoM$_8$

> I carry a heavy workload: 18–30 teaching hours per week for us full-time! I got used to doing this face-to-face. When I shifted online, I felt like I was in a different world. Everything changed—my pedagogy, my time zone, my life. —MoM$_7$

Our academic mothers were overwhelmed by complex technology for online pedagogy. Needless to say, the heavy workload increase was due to the shift to online teaching as well as the border crossing between work and home. Increased attention to students' needs during online teaching added to the already taxing roles of these academic mothers. They wondered if their students listen to them during synchronous classes. The home environment was not conducive to a healthy online discussion free from distraction. Two academic mothers described their experience:

> Some of my students attend my class when they have just awoken from bed. They are all anxious, depressed, worried, and lost. —MoM$_6$

> I wonder if students are listening to me when their camera is off. When the class is over and they have to log out, some students are left logged in for long; they might have fallen asleep. —MoM$_2$

Two of the academic mothers I interviewed are single parents who have kept everything up in the air; their marriage did not work even in the early stage of their married life. Now, the challenges brought about by this pandemic were taking a toll on them. Here are their short narratives:

> I have been separated for five years. My husband doesn't support my two schooling children. Last year, my daughter attempted suicide three times. I saw the slashes on her wrist. I was devastated. I lost my purpose. I lost myself. I am a failure. —MoM$_2$

> I feel that raising my children all by myself has its ups and downs. I can't say they would have been better off if their father had been around; they might not

even be. I just want to survive. I need to regain myself; I'm drifting away from myself. —MoM$_4$

One academic mother could not make sense of reality. Her enormous fears are caused by financial resources getting scarce due to the increasing medical costs of her bedridden husband. There were times she wanted everything to just end.

I am tired. And I am enraged, in fact—most of the time. My husband has been paralyzed for three years now. How can I sustain his medication? He tested positive last year. I got so anxious. Where would I get the money for the hospitalization when we don't even have enough for our daily expenses? I'm tired of thinking about what will happen in the future. I feel lost! —MoM$_1$

From the previous discussion, I interpretively summarized the key findings on how academic mothers suffered enormously in balancing their work–family role demands during the first few months of 2020. The challenging changes that warped home and workspaces and their shifting finances brought unimaginable uncertainties to them. Coupled with the changing challenges of stress-filled virtual teaching and the gripping anxieties of getting sick, these academic mothers had their household budgets utterly depleted. These were the challenges for which they were not at all prepared. Some academic mothers even tried to share with their husbands how they felt they were ignored by them; while some husbands were open to listening to them, others did not take it well. This growing indifference to their plight added insult to injury when all they wanted was to be held, heard, and helped.

Yet, despite all these challenges, these academic mothers navigated and survived the changes and challenges in their lives. They were made of hard stuff.

*5.2. Championed Choices*

A *choice championed* is a crucial decision an individual vigorously and rigorously makes against all odds to ensure that all people under their care stay afloat. It is making critically important choices as they deal with the inevitable. The eight academic mothers interviewed are tough women who kept body and soul together. Surely, this pandemic disturbed the rhythm of their actual lives. While these real-life changes stretched them too thin, the hard choices they made led them to choose to fight when no one else thought of fighting for them.

Two sub-themes were elicited from the second theme: (i) feminine grit, and (ii) feminine wit. The first sub-theme showed the strength of the academic mothers amid all the challenges they went through; the second sub-theme referred to the woman's quick and smart response to a disruption or to something unexpected under constraints.

5.2.1. Feminine Grit

Angela Duckworth, who grew popular because of her viral TED talk on grit, defined the word in her book, *Grit: The Power of Passion and Perseverance*, as "a passion to accomplish a particular top-level goal and the perseverance to follow through" (Duckworth 2016). While her concept of grit applies to all regardless of gender, I tried to feminize the term to make it exclusive to women, hence, the term *feminine grit*. What we can say about academic mothers is that they are both tough and soft, their worth abysmally unrecognized, and their loss appallingly costly. Amid this disturbing and disruptive pandemic, these academic mothers, displaying their feminine grit, showing determination, courage, and persistence, chose to persevere and befriend uncertainty by staying gritty and fearless. Here are some of their gritty lines:

Why am I strong? Academic mothers in the Philippines are more capable of adjusting because of the series of disasters happening in our country—typhoons, earthquakes, and diseases. We already know how to react to all of these disasters. The COVID-19 pandemic is just one of those. I am already immune. This COVID is just one of those catastrophes. I am a survivor. —MoM$_4$

One mother mentioned that she could weather all storms in her life as long as she has her family to back her up:

> I let go of what I cannot control. Embrace life's uncertainties. Don't allow yourself to be overwhelmed. Do things one at a time. Stop romanticizing pains. I've been through a lot, and I still can take anything that happens for as long as I am with my family. —MoM$_7$

It was not an option for academic mothers but an existential necessity to be strong for their families. This show of a brave front is an instantiation of liminality, a posture of invulnerability, shouting with silenced voices their firm resolve that they would emerge victoriously unscathed after this great storm in their lives.

> Mother to three children, I managed to weather all the challenges that came into my life. You never know what I've gone through. I am used to all pains; I can face them all for my family. I am a tough woman. —MoM$_8$

The many adversities these academic mothers suffered in life made them better at handling challenging events because of their feminine grit. These women felt more responsible than their male spouses for sustaining relationships at work and home. They managed to provide robust support in one domain without impairing the other in order not to disrupt the harmony in the two gendered spaces—home and work. Their male spouses did not feel obligated to do the same.

### 5.2.2. Feminine Wit

Feminine wit, a phrase I coined, is the willingness to own one's choices—a difficult willingness—that promises the corrective to existential helplessness, vulnerability, and agony. It pertains to the woman's ability to gather her wits calmly and clearly under such demanding circumstances. Further, to possess feminine wit is to lead an inspiring and inspired life, making life's trials a training ground for character and virtue. The academic mothers' insightful message is a treasure trove of penetrating thoughts about how they played their strength in making choices. Pained by life's vagaries, they stayed witty and gritty for their families. Gritty, they chose to fight. Wounded, they chose to heal. Witty, they chose their future with their present action.

> Life is tough. But it won't wait for you when you feel better. You need to choose to make life better. Life will not make it better for you. I need to choose to be witty. I need to choose me. —MoM$_2$

> The eldest among my siblings, I am tasked to care for my sick father. The "rock" in the family, I take care of my husband and my children during this pandemic. I was raised strongly by my mother. Standing tall, that's the only option I have. —MoM$_3$

The work they performed during this pandemic should be unequivocally valued because they developed a skill set in navigating and negotiating two very challenging spaces—the home and academia.

> We could choose to feel overwhelmed to the point of inaction or choose to fight it out with all our vigor and wit. I chose the latter. And I am winning this battle. I involve my children in whatever I do—my work, social life, and personal life; I want them to know what's happening and how I handle the situation. They know what's happening to me, and they fully support me. I want them to tell their own children someday my story. —MoM$_4$

> The first battle is always staged in mind. And we have to conquer our fears, anxieties, doubts, and concerns. If we are able to transcend all this, half of the battle is won. And that's precisely what I did. I faced the battle head-on; I still do. —MoM$_8$

As revealed by the preceding narratives, academic mothers' wit and grit combined make them perform better despite life's sharp vicissitudes. At times, some situations

presented insurmountable challenges to these academic mothers, such as an important meeting and their child's recognition ceremony or an invitation to present a research paper at a conference and a paralyzed husband needing to be hospitalized happening at the same time—one of these two priorities unavoidably suffered, not because she is an awful mother or a terrible professor, but because life does not always come in tidy packages. There is nothing dramatic in the event, and there is no need to romanticize the situation. They just needed to be gritty and witty.

## 6. Discussion

Undoubtedly, the lives of academic mothers changed sharply due to the COVID-19 pandemic. Overwhelmed by their daily tasks, they all agreed that the pandemic unexpectedly upended their lives. After global stay-at-home school orders shifted pedagogy to emergency online learning, the work–life balance for academic mothers tilted in their disfavor. They suffered tremendously during the first few months of the year 2020; two years into the pandemic, they persisted in being over-gritty. Given these unforeseen and unimaginable new factors in their lives of mothering and teaching during the pandemic, a plethora of anxieties daunted, haunted, and taunted them to no end.

In the Philippines, the situation is no different. The current study explored the multiple challenges college academic mothers experienced in navigating their embodied maternal lives in a disembodied academic world. In particular, it investigated how they balanced the challenges of contemporary motherhood and academia during the pandemic. The findings revealed that this COVID-19 pandemic imperiled our academic mothers tremendously. The study by (Fulweiler et al. 2021) aligned with these findings, discussed the causes of stress among academic mothers: the new work-from-home setup, financial setbacks, and the shift to an online mode of teaching. The eight academic mothers I interviewed were plagued with enormous challenges during the pandemic; yet, they were able to manage to integrate work and family the best way they knew how. I heard their oscillating stories of guilt, grief, and grit. To make sense of their narratives, I used Levinas's framework of unlimited responsibility in shedding light on maternal self-sacrifice and David Whyte's work–family empowerment in theorizing work–family balance.

Those familiar with Levinas will recognize that this scenario brought to mind certain aspects of his writing on ethical maternity that sheds light on the sacrifice a mother is willing to make for others. Drawing from Levinas' framework of disinterested responsibility, I used motherhood as an embodiment of responsibility and self-sacrifice: a mother's single-minded, wholehearted, ultimate selflessness displayed through her maternal, fleshy body. The unselfish effort arouses in a mother a joy more sublime and purer than the delight of personal enjoyment. Here we understand that no matter how difficult life offers to a mother, she is willing to give it all as an articulation of motherhood that remains in the flesh. Thrust into the strange and alien world of the pandemic, the pressing needs of her family unsettle her more, forcing her to give more, serve more, love more—all for her family (Espartinez 2014). Levinas insisted that any ethically responsible relationship should be governed only by the language of selfless service to others (Espartinez 2021, p. 3).

Transitioning from "work self" to "home self," academic mothers revealed some cracks in managing academic and household tasks. Enduring struggles to reconcile professional and maternal domains in warped spaces showed how shifting blurred territories became so confusing that their vulnerabilities were deeply exposed. Further, despite the shift in financial roles, where women turned out as breadwinners, academic mothers devoted more time spent on childcare and household work (Mohamad and Despois 2022). They still assumed the traditionally gendered roles of caregiver and homemaker at home to meet the expectations of their feminine duties. Despite their double burden, these academic mothers survived the trials with their hands calloused from household chores, their minds perturbed by challenging academic anxieties, and their hearts pained by the thought of being "*not enough*." This feeling of not-being-enoughness revealed to us something important. When

we call their reaction overacting or dramatics, what we mean is that these academic mothers turned out stronger than we thought they would be.

For these academic mothers who tried to make sense of their changed and challenged lives, it is not uncommon to feel that life was too much. Their gripping trepidations, the feeling of losing one's focus due to overwhelming pressures and stressors that the pandemic brought to their lives, cripplingly surprised them. Moreover, it is not uncommon to feel the expectation that they really should be up to it; there may be too many demands and too little supply of time, too more for "others" and too less for "myself" during the pandemic. Challenged with the stresses and strains of quotidian life, these academic mothers felt that they were failing and not up to the task.

What is more revealing—and even more surprising—is that these academic mothers accepted their role of being the family's breadwinners and homemakers to equal degrees and were even hesitant in admitting it as evidenced by their narratives. They justified this unspoken inequality in parental duties as acceptable, unavoidable, and endurable. The reversal of financial roles that led to the loss of breadwinner status contributed to man's belligerent conduct at home (Myers and Demantas 2016) as they were viewed to be less masculine (Springer 2010). This is what one academic mother in our study has been experiencing with her hegemonic paralyzed husband. Studies showed that many husbands maintained their gender privilege in the home even after losing work. Legerski and Cornwall (2010) and Hochschild and Machung (2012) discovered in their study that when men "breadlose," they refused to take part in household work in an attempt to reinforce their masculinity. Further, if they opted to dip their hands in household chores, they enjoyed some latitude over the degree to which they choose to do housework (Ashforth et al. 2000). These findings revealed, to a significant extent, that something more is needed than merely problematizing feminine breadwinning and male breadlosing; the vulnerabilities that academic mothers from the social construction and performance of "female breadwinning" endured disproportionately must also be explored.

I interviewed these eight academic women for an hour of brilliance, bravery, and beauty, bringing to the dialogue their narratives of defeats and victories to make it immensely stimulating and profoundly balanced. This interview revealed that, given their feminine wit and grit, our academic mothers were able to surpass all the challenges of contemporary motherhood and academia during the pandemic. They chose to fight their battles, and they were winning. They courageously displayed their feminine capacity to thrive after adverse events. With a shared world, these academic mothers celebrated the triumph of the human spirit and their feminine wit expressed in their thoughtful sensitivity and humanity. Their deeply unsettling narratives will continually daunt, haunt, and taunt us with their vulnerability, vigor, and valor.

In his philosophical book, *The Three Marriages: Reimagining Work, Self and Relationship*, David Whyte challenged the conventional concept of work–life balance (Whyte 2009). Work–life balance, Whyte (2009) argued, "is, at its heart, nonnegotiable" (p. 17) and invites us to view each not as a tug-of-war pitting the professional and personal against the other but as something "conversing with, questioning or emboldening the other" (p. 18). Work and life are not at war, vying for supremacy or control; instead, as Whyte further claimed, they are best viewed as a "movable conversational frontier." Academic mothers' work and family should blend well together, not compete against each other. It is not a matter of balance as much as one of empowerment and enablement. While work is undoubtedly a part of their lives, it is not a separate and distinct part of life's tasks. The interplay of work and life should not be viewed as a burden but as something that makes us be; it needs us as much as we need it. It does not make compromises with our lives. Rather, work and life feed each other.

Work and home, for these academic mothers, are in a constant conversation, the back-and-forth dialogue between what the world needs of them and what they need of it, what they can do and what they cannot do, who they know they are, and who they know they are not. They negotiate between what they can offer life and what life can offer

them. They are capable of constantly surprising and dislodging themselves with their challenges and how they reconstruct themselves and how much grit, fortitude and maturity this pandemic made of them. Whyte invites us to normalize work–family conversations in a continuous, non-threatening manner, letting work and life seesaw between fits of delight and impetuousness. Here, I advise, following Whyte, that instead of silencing women's voices, we should allow the surfacing of the discourse of work–family narratives by inviting academic mothers to narrate their own stories of grief and grit.

Care work is a noble task and should be made a respectable occupation. However, the caregiving burden of mothers in unpaid care work at home becomes challenging during the pandemic (Akuoko et al. 2021; Boesch and Hamm 2020; Del Boca et al. 2020; Hupkau and Petrongolo 2020) when husbands view care work as a second-rate, worth-less job. Women are more caring, but whether framed as a barrier or an advantage, these beliefs hold women back. Deeply rooted gender stereotypes dictate that men do the breadwinning and view unpaid caring responsibilities as a less-important female job. Similarly, entrenched social norms release men from taking up care work at home. Even in gender-atypical parents (at-home fathers and breadwinning mothers), gendered housework specialization remains at work despite the mother's reduced time availability and breadwinning role (Akanle and Nwaobiala 2020; Chesley 2011, 2017; Legerski and Cornwall 2010). No bargaining power for reduced domestic involvement was given to these academic mothers. The gender-normative work–family arrangement remained at play.

By privatizing and silencing the emotions of the academic mothers, the hegemonic narratives have ensured that the marginalized—the academic mothers for our purpose in this study—cannot name their marginality. Unnamed, their subaltern stories have been kept hidden deep and their recurrence—always muted and subdued—further strengthened their staying at the margins. It is also visible in the occlusion of feminine discourses that lie outside the dominant hegemonic ones that are considered official, authentic, and normal. Instead of silencing academic mothers' voices, we should rather normalize the discourse of work–family narratives by inviting academic mothers to let their subaltern stories surface, especially those pertaining to their lived experiences of subordination and subalternity and challenge hegemonic or dominant assumptions about work–life balance. As long as we perpetuate these traditional roles as hierarchical rather than complementary, academic mothers will unnecessarily bear the brunt of care work. Unsurprisingly and sadly, some wives tolerate their husbands, even preserving the male breadwinner illusion (Sánchez-Mira 2021).

We need to challenge this dominant notion of alpha male culture in the family that normally creates a "patriarchal dividend that privileges men over women" (Lindegger and Quayle 2009) and start to "break the silence." One of the biggest risks that silenced marginality carries is the possibility of their own daughters becoming like them and furthering the cycle of stigmatized professional mothers. This can occur across several generations, with each accruing unresolved burdens for the next. Academic mothers can begin to care from a space of choice and love, not duty and dread of abandonment. They should stop narrating the hero myth. Uncorrected, the cycle will go on ad infinitum. It has to stop. *Now*.

In more than an hour of an interview with these academic mothers, I recognized their carefully masked perturbation, trying to show a brave front as they attempted to make their maternal strength indiscernibly impaired. When mothers are more accepting of male privilege and care work is still considered women's duties, these women empower people around them to define to them what they fail to define themselves. I am worried that as these women survive and thrive amidst the pandemic challenges, their capacity for resilience becomes more of an act of subconscious protest. Two years into the pandemic, college academic mothers in the Philippines are finally getting the hang of this work-from-home arrangement, and they are now enjoying every bit of it such that they did not want to go back to their physical classrooms. What once presented itself as affliction now turns into affection, a bane turned into a boon, and the curse of the pandemic turned into a blessing.

Needless to say that the pandemic served a physical and mental feat for women—all the more for academic mothers. But these women academicians made it through the rain. They survived the difficulties that life threw at them. Are these Filipina college professors really strong, or just playing strong so that all seems well with them? If they made it, it does not mean the challenge was not a feat; it was because they managed to weather the challenges for their family, not to mention some husbands that refused to step up. Sure, they needed to enlist their husbands for help, which usually did not come when needed the most. But with or without a man, these women still performed at the top of their game.

This study invites scholars, school administrators, and policymakers to take a deeper look at the challenging role of academic mothers in the Philippines as they perform their interroles as teachers and homemakers and the prospects for adjusting university priorities, policies, and practices in these times of changing challenges and challenging changes during and even beyond the pandemic. Academic policies and government support that offer flexible and reduced academic work for our academic mothers would be a welcome academic move. Unless policies are made flexible to suit the academic mothers' circumstances, social and moral inequities persist affecting our faculty mothers.

## 7. Conclusions

From the preceding discussion, I argued that mothering and professing are sites both of silent contestation and of inner triumph. From the multiple challenges that academic mothers in the Philippines went through, they emerged relatively victorious with their feminine grit and wit. They were able to navigate and accomplish the work in two very challenging gendered spaces—the home and academia. I arrived at one of the most profound ongoing threads of this feminine interview. It revealed a contemporary poignancy—the question of personal guilt, family devotion, and maternal responsibility—and the crucial resolve of these academic mothers in ensuring and assuring a constructive, hopeful, and optimistic path forward.

From this study, I conclude by viewing work and family not as inherently conflicting and mutually exclusive but as allies allowing the continuous conversation between work and family to surface. To attain this enabling and empowering integration, there is a need to bring the academic mothers' personal histories into discussions to foreground their maternal vulnerabilities on the one hand and their maternal triumphs on the other hand.

To end, I register my three humble pleas for the sake of these academic mothers. *One*, I hope that they continue to sustain the group of personal and professional support that they have. These are constructive spaces, offering women opportunities to build coping strategies and support one another to sustain their efforts to combine work and family. *Two*, I call for their male partners to share domestic commitments *equitably*. For sure, they can find it in their *hearts*—and more so in their *hands*—to lift a finger and split the dishes and laundry clothes. Finally, *third*, robust institutional and government support that offers a flexible and reduced academic work environment for our academic mothers would greatly help. Tilting educational policy toward unburdening women faculty members may include reducing teaching six courses to four courses per semester as practiced in most foreign universities.

When this pandemic is over, and even if another challenge comes along, these academic mothers reappear better than before, with their work and family life becoming one ecstatic mélange of a happy existence. May this study serve as a platform to disrupt the gender structure in ways that are hospitable to women who are making sense of work–family enrichment.

**Funding:** This research is supported by the Faculty Research Program of De La Salle-College of Saint Benilde with Grant Code FRP-12152021-C-06122022.

**Institutional Review Board Statement:** This study was conducted after obtaining approval from the Institutional Review Board of Benilde-ARRC with Certificate Reference #11162022-FRP-001.

**Informed Consent Statement:** Informed consent was obtained from the research participants to conduct the study; consent was likewise given for its publication.

**Acknowledgments:** I acknowledge the support of the Office of Institutional Effectiveness and Research for its Faculty Research Program Grant awarded to me. I am also greatly indebted to all the anonymous referees who helped me in the reviewing process by providing valuable insights, comments, and suggestions for the improvement of this article.

**Conflicts of Interest:** The author declares no conflict of interest.

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
