# Peer review of "Lived Experiences of Mothering and Teaching during the Pandemic: A Narrative Inquiry on College Faculty Mothers in the Philippines"

_socsci, doi:10.3390/socsci12010024_

Round 1

Reviewer 1 Report

In this qualitative study of 8 mother college professors in the Philippines, the author analyzes interview data about how mothers cared for family members while working as academics during the COVID 19 pandemic. Drawing on literature about working women in academia, the author designed the study to include academic mothers who were caring for children aged 18 and under during the pandemic. The author organized the findings around 3 themes: challenging changes, changing challenges, and championed choices and suggests implications for strengthening both theories of carework and policies for work-life balance.

The literature review, recruitment strategy, analysis, findings and recommendations all are couched within heteronormativity. The author should be explicit that she is limiting the entire study and article to heterosexual women who have (or have had) men partners. The privileging of these experiences without acknowledgement is problematic in terms of diversity, equity and inclusion.

The findings need further contextualization beyond listing series of quotes. The findings are rich and should be better unpacked and analyzed rather than just including multiple excerpts illustrating each theme. Within each section of quotes are multiple findings that are not addressed or discussed; rather they are combined and lumped together within a broad theme. Each quote should be situated within the theme with commentary and analysis to strengthen the claims being made and quotes should be connected through the author’s narrative.

The discussion section includes a lot of new themes from the literature. It is not clear whether the themes were actually found in this study, or if the literature is expanding on the findings. This is confusing. For example, under “normalizing work-family narratives,” the author discusses how the loss of breadwinner status contributes to men’s belligerent conduct at home and is emasculating. How many of the participants in the study experienced this? How does it relate to this study and its findings and implications? Why is this being included in the discussion?

The verb tense changes throughout and needs to be consistent.

Author Response

SUMMARY OF REVISIONS (Reviewer 1)

Title                                  Lived Experiences of Mothering and Teaching During the Pandemic: A Narrative Inquiry on College Faculty Mothers in the Philippines

REMARKS AND SUGGESTIONS

REVISIONS

In this qualitative study of 8 mother college professors in the Philippines, the author analyzes interview data about how mothers cared for family members while working as academics during the COVID 19 pandemic. Drawing on literature about working women in academia, the author designed the study to include academic mothers who were caring for children aged 18 and under during the pandemic. The author organized the findings around 3 themes: challenging changes, changing challenges, and championed choices and suggests implications for strengthening both theories of carework and policies for work-life balance.

This is correct.

The literature review, recruitment strategy, analysis, findings and recommendations all are couched within heteronormativity. The author should be explicit that she is limiting the entire study and article to heterosexual women who have (or have had) men partners. The privileging of these experiences without acknowledgement is problematic in terms of diversity, equity and inclusion.

This is a very good comment.  Given this suggestion, the revision is stated this way: the study is couched within heteronormative marital relations as the Philippines still favors opposite-gendered marital unions as the default sexual relationship. This explanation is included in lines 44-43 of the revised manuscript. Also in lines 86-88, “The study articulated the complexities of motherhood and the teaching career of eight academic mothers in opposite-gendered marital relationships in Philippine tertiary education.”

The findings need further contextualization beyond listing series of quotes. The findings are rich and should be better unpacked and analyzed rather than just including multiple excerpts illustrating each theme.

Within each section of quotes are multiple findings that are not addressed or discussed; rather they are combined and lumped together within a broad theme.

Each quote should be situated within the theme with commentary and analysis to strengthen the claims being made and quotes should be connected through the author’s narrative.

A major revamp was made to respond to your constructive suggestion. The changes even made my presentation airtight. To unpack the rather rich findings of the study, the following revisions were made:

·       the quotes are now explained in more detail in the Results, weaving them together by linking them within the broader theme.

·       A better and clearer discussion of the findings is now presented in the Discussion section to see the connection of the quoted lines with the emergent themes.

The discussion section includes a lot of new themes from the literature. It is not clear whether the themes were actually found in this study, or if the literature is expanding on the findings. This is confusing.

For example, under “normalizing work-family narratives,” the author discusses how the loss of breadwinner status contributes to men’s belligerent conduct at home and is emasculating. How many of the participants in the study experienced this? How does it relate to this study and its findings and implications? Why is this being included in the discussion?

Section Headings previously presented in the Discussion were deleted to avoid confusion as they might suggest new themes that were not included in the Results. Further, items that were not included in the Results were likewise deleted from the Discussion.

This section is renamed Shifting Gender Roles: Female Breadwinning, Male Breadlosing to justify the inclusion of this line: The reversal of financial roles that led to the loss of breadwinner status contributed to man’s belligerent conduct at home (K. Myers & Demantas, 2016) as they were viewed to be less masculine (Springer, 2010) found in 548-550.

With these revisions, the arguments and discussion of findings are made coherent, balanced and compelling The conclusion is fully supported by the research findings.

The verb tense changes throughout and needs to be consistent.

The verb tense is made consistent all throughout the paper with the assistance of a proofreader.

Reviewer 2 Report

This article presents a necessary, original and current topic. The work is well founded.

It is only recommended that the general objective be revised. It should be concrete and measurable.

Author Response

SUMMARY OF REVISIONS (Reviewer 2)

Title                                  Lived Experiences of Mothering and Teaching During the Pandemic: A Narrative Inquiry on College Faculty Mothers in the Philippines

REMARKS AND SUGGESTIONS

REVISIONS

This article presents a necessary, original and current topic. The work is well founded.

Thank you.

It is only recommended that the general objective be revised. It should be concrete and measurable.

This qualitative research focuses on gathering mainly narrative information rather than empirical data. Gathered qualitative information is then analyzed in an interpretative manner.

Some other revisions were incorporated. With these revisions, the arguments and discussion of findings are made coherent, balanced and compelling The conclusion is fully supported by the research findings.

Reviewer 3 Report

I really like the outline of the paper and the topic. I think it is very important to address. I have some suggestions to improve the paper in my view.

First, I would prefer if the literature review is more focused on the topic about motherhood in academia and precisly on the newer studies which have been conducted since the pandemic. In the introduction there are already some mentioned, but in the literature review the paragraph about it is very short and raises more questions. Here it would be great to give an overview about the already conducted studies about academic motherhood in the pandemic: there are central findings, for example mothers have been less access to research funding, have less published, less time for research and have been faced more with the troubles of remote teaching as many women are employed as lecturers. Then the author had the oppoortunity to place its paper into this state of the art. Until today there have been a focus on studies in US and Europe. THe paper has novelty because it shows how teaching women in Philippines dealed with the problems.

Second, this would give more instructive categories for the presentation of the findings. I think this chapter could be improved. Instead of presenting one quote after the next, I would prefer if the presented quotes would be linked with the existing state of the art and its gaps. For example is in the state of the art already mentioned that academic mothers have been more affected by online teaching, this could be linked to the finding of the "technostress online teaching". It is more or less a bit abritrary or sprawling what is presented in the findings. Despite that in the discussion the author combines the findings into three main aspects, which is convincable. 

Author Response

SUMMARY OF REVISIONS (Reviewer 3)

Title                                  Lived Experiences of Mothering and Teaching During the Pandemic: A Narrative Inquiry on College Faculty Mothers in the Philippines

REMARKS AND SUGGESTIONS

REVISIONS

I really like the outline of the paper and the topic. I think it is very important to address. I have some suggestions to improve the paper in my view.

Thank you.

First, I would prefer if the literature review is more focused on the topic about motherhood in academia and precisely on the newer studies which have been conducted since the pandemic.

In the introduction there are already some mentioned, but in the literature review the paragraph about it is very short and raises more questions. Here it would be great to give an overview about the already conducted studies about academic motherhood in the pandemic: there are central findings, for example mothers have been less access to research funding, have less published, less time for research and have been faced more with the troubles of remote teaching as many women are employed as lecturers. Then the author had the opportunity to place its paper into this state of the art. Until today there have been a focus on studies in US and Europe. The paper has novelty because it shows how teaching women in Philippines dealed with the problems.

The literature review is presented using a funnel technique which I find useful in starting from generalities to the specific focus of the paper. At the top of the funnel, I am farthest from the problem (Women’s Participation in the Workplace) and at the bottom, I am closest to the problem (motherhood in academia).

Since my study is on motherhood and academia, my literature review focused more on the studies centering around these topics. As the study ended in March 2022, only studies made up to this period was included in the review. However, other studies such as issues on reduced scientific productivity, less access to research funding, less time to publish and to do research are now incorporated to update the literature review. They are in lines 125-127 and 162-163 of the revised manuscript:  They published less and suffered reduced research productivity (Armenti, 2004b; Minello, 2020; Minello et al., 2021; Mohamad & Despois, 2022; Orchard et al., 2022). Challenges in remote teaching (literature revie updated) are likewise included.

Second, this would give more instructive categories for the presentation of the findings. I think this chapter could be improved. Instead of presenting one quote after the next, I would prefer if the presented quotes would be linked with the existing state of the art and its gaps.

For example is in the state of the art already mentioned that academic mothers have been more affected by online teaching, this could be linked to the finding of the "technostress online teaching". It is more or less a bit arbitrary or sprawling what is presented in the findings.

Quoted lines were neatly interwoven through meta-narratives to make meaningful connections with the broader theme.

A major revamp was made to respond to your constructive suggestion. The changes even made my presentation airtight. To unpack the rather rich findings of the study, the following revisions were made:

·       the quotes are now explained in more detail in the Results, weaving them together by linking them within the broader theme.

·       A better and clearer discussion of the findings is now presented in the Discussion section to see the connection of the quoted lines with the emergent themes.

The comments are addressed by linking  technostress in virtual teaching with the broader theme: 5.1.3 Shifting Fears

 (the Discussion are now incorporated in lines 381-412. The highlight of the revision is this:  Academic mothers transitioning to online learning needed to learn how students drop ideas into the chat box, quickly answer questions via online polls, and do small-group discussions in virtual breakout rooms. All these technologies were new to them. While face-to-face teaching and learning was already a huge hurdle, the difficulty is more compounded by the online teaching mode (5.1.3.1 Fear of Complex Technology in Online Teaching) found in lines 381-401.

Despite that in the discussion the author combines the findings into three main aspects, which is convincable.

Thank you very much.

Round 2

Reviewer 1 Report

This revision is improved over the original submission. What I think still needs a strengthening is an extended discussion of the theoretical framing of the study within embodiment. The author uses that term without defining it or citing the particular understanding. There is a vast literature on embodiment and the author needs to situate their study within that literature. How is motherhood embodied, and academia disembodied? The author states this without elaboration or example of what this means.  

There are also several grammatical and writing errors throughout the manuscript.

Author Response

REMARKS AND SUGGESTIONS

REVISIONS

This revision is improved over the original submission. What I think still needs a strengthening is an extended discussion of the theoretical framing of the study within embodiment. The author uses that term without defining it or citing the particular understanding. There is a vast literature on embodiment and the author needs to situate their study within that literature. How is motherhood embodied, and academia disembodied? The author states this without elaboration or example of what this means.  

There are also several grammatical and writing errors throughout the manuscript.

Embodied motherhood and disembodied academic life is also clarified in lines 123-135.

Theoretical framing within maternal embodiment as contrasted with disembodied academic life is included in the Discussion lines 592-594. 

Further proofreading is done to resolve grammatical and writing errors. Consistency of verb tense is observed.

Reviewer 3 Report

the state of the art has now several undersubheadings, I would suggest to make 2.1, 2.2, 2.3 ... instead of 2.1, 2.1.1., 2.1.1.1

the chapter about the findings has improved a lot, please check if there are more links between the existing literature and your findings. Which are induced by the pandemic? Which by the local context (country) ?

Author Response

REMARKS AND SUGGESTIONS

REVISIONS

the state of the art has now several undersubheadings, I would suggest to make 2.1, 2.2, 2.3 ... instead of 2.1, 2.1.1., 2.1.1.1

the chapter about the findings has improved a lot,

please check if there are more links between the existing literature and your findings. Which are induced by the pandemic? Which by the local context (country) ?

Cumbersome numbering in sub-headings are now removed without sacrificing order and clarity. 

Other links were included (lines 55-58) by country.